# The "Social" Side of Big Data: Teaching BD Analytics to Political Science Students

**Giampiero Giacomello ***  **and Oltion Preka**

Department of Political and Social Sciences, University of Bologna, 40125 Bologna, Italy; oltion.preka@unibo.it
* Correspondence: giampiero.giacomello@unibo.it; Tel.: +39-051-206-2668

**Abstract:** In an increasingly technology-dependent world, it is not surprising that STEM (Science, Technology, Engineering, and Mathematics) graduates are in high demand. This state of affairs, however, has made the public overlook the case that not only computing and artificial intelligence are naturally interdisciplinary, but that a huge portion of generated data comes from human–computer interactions, thus they are social in character and nature. Hence, social science practitioners should be in demand too, but this does not seem the case. One of the reasons for such a situation is that political and social science departments worldwide tend to remain in their "comfort zone" and see their disciplines quite traditionally, but by doing so they cut themselves off from many positions today. The authors believed that these conditions should and could be changed and thus in a few years created a specifically tailored course for students in Political Science. This paper examines the experience of the last year of such a program, which, after several tweaks and adjustments, is now fully operational. The results and students' appreciation are quite remarkable. Hence the authors considered the experience was worth sharing, so that colleagues in social and political science departments may feel encouraged to follow and replicate such an example.

**Keywords:** social big data; teaching Python; Jupyter Notebook; Pandas; Matplotlib

## 1. Introduction

The technological advancements observed over the last decade now affect all aspects of our lives. One of such evolutions is the quantity of new data produced every day: huge amounts of data are being generated every second from multiple sources, including electronic and, more generally, digital devices. Social media interactions or any other kind of tracked activity, together with the availability of digital documents, represent examples of data sources that did not exist 20 years ago. Getting access to such data lakes has opened new opportunities for social scientists changing some of the paradigms of the past. If properly treated and analyzed, such data can be a powerful instrument of information to improve our knowledge about specific arguments of interest in social sciences.

One of the traditional missions of sociologists and political scientists as social scientists has been to explain what happens when new elements, such as huge volumes of data, impact upon societies. Thanks to the "wealth" provided by Big Data (BD), and Artificial Intelligence (AI) as applied to social research, such as in the legal sector, in electoral studies, political communications or in defense, analysis is exploding. It is, however, in the availability and possibility of analyzing much larger quantities of data, that lies the path to better methodology in the social sciences, which is, incidentally, the only alternative for disciplines that cannot rely extensively on the experimental method.

Undoubtedly, this is a *terra incognita*, a completely unexplored realm of knowledge transmission for information and computer scientists, but for political scientists too; the most important journal dedicated to education in Political Science (PS), namely the *Journal of Political Science Education*, in the last three years (2017–2019) has published only one article specifically on "data analysis and

visualization" [1]. There are, obviously, articles here and there in political science subfields, particularly for text analysis [2] or even for teaching R and Big Data [3], but these are rare. In sub- fields such as Public Policy/Administration or research on voting behavior and political parties, teaching quantitative methods has always been central, but these specializations are more the exception than the rule in the Political Science domain. In general, many political scientists are more "reluctant" when it comes to teaching data analytics than other social scientists such as sociologists and psychologists, not to mention economists.

All the more so, it is critical to publish and provide a wider audience with solutions to this problem. Indeed, the Special Issue on "Human Computer Interaction in Education" of *Multimedia Technologies and Interaction* in 2018 is not only indicative of the interests there are for the sharing of teaching experiences among colleagues and instructors, but it is also an example to be replicated for specific disciplines.

Computer and computer networks are no strangers in the social sciences of course, and indeed there has been a much larger interest for them by social/political scientists, especially after the advent of the internet. The field of communications has, of course, been at the forefront e.g., [4], followed by sociology [5], thanks to the overall interests in networked societies with the work of, for example, Manuel Castells [6], and by the mass of data produced by social media e.g., [7,8]. The obvious international dimension of computer networks has attracted international relations scholars e.g., [9]. Nevertheless, focusing on "cyberspace" (instead of the "international system" of old), the debate in international relations has remained quite limited, as far as within the "technical" dimension is concerned e.g., [10].

In addition to the comprehensible expectations for the positive consequences that Big Data will have on the various fields of social sciences [11], there are currently at least two principal disciplines that try to merge computing and the social sciences, namely computational social science [12] and the more recent (thanks to social media data) social computing [13]. The teaching experience examined in this paper belongs to the latter discipline, as the authors think that, when it comes to entering the job market, Political Science students may be more competitive in those skills related to social media and human-computer interactions.

Despite the impressive and wide-ranging progresses made in research, it is rather difficult for the private sector and businesses to understand the relevance and importance of those that are trained in such disciplines. It is not like the situation with graduates in STEM (Science, Technology, Engineering and Mathematics), as social sciences graduates, and in particular those in political science, face an uphill struggle when it comes to be hired for jobs that, in growing number, require some dexterity with computing and coding. Such students may know what the significance and roles of Big Data, computing, and AI is for advanced societies, but they often fail to make that knowledge appreciated by their employers because there is a gap between that "birds-eye view" comprehension and how to translate it into analytical advantage for their potential employers. Convinced that closing that gap is doable and easier than it appears to many colleagues, we decided, with another like- minded colleague, to embark in a (now) five-year old teaching experiment.

The first four years have been a long trial-and-error process, with further adjustments and tweaks. Then in the academic year 2018/2019, the authors concluded that the teaching experiment had moved on from the early stage and was now "mature" so that it could be replicated elsewhere. To provide a usable "blueprint" for Big Data for such purposes has been the main reason to prepare this paper. The next sections will illustrate the teaching experience in detail with the results of students' evaluations and in-house survey. Clearly, the adjusting process will continue and we, as "reporters" of this experience, hope that colleagues from other colleges and universities will enrich the knowledge base with their own experience. In the next section of the paper, it is probably more convenient for the reader to be first exposed to the methodology used, before presenting the outcome of the experience, which will follow after the section on methodology.

## 2. Materials and Methods

This section describes our experience in teaching the class "Big Data for Social Sciences", an optional course part of a Political Science degree (M.A. in International Relations and M.A. in Public Corporate Communication) at the University of Bologna for the academic year 2018/2019. The process of changing and adapting, however, began in the academic year 2015/2016 with a brand-new course on Big Data analysis.

The main objective of the course was to provide students with fundamental tools, skills, and methods that enable them to apply data science techniques in the context of social sciences, and to contribute to the "social" understanding of computing in our societies. With the purpose of lowering entry barriers for students with a social science background, the entire course was designed as a top-down approach with a major focus on hands-on skills that help students lever Big Data. The software tool adopted in this course was Python with Anaconda distribution, and it was combined with Jupyter Notebook platform. More specifically, by the end of the course, students were expected to:

1.  have developed coding skills at a medium level in Python, and be confident with the Python ecosystem, including its most commonly used libraries: Pandas for data analysis, Matplotlib for data visualization, and NLTK for text analysis and mining techniques;
2.  have learned how to research sources of data and be familiar with the main public data sets available;
3.  have acquired the main methods and techniques to carry out Big Data analysis, both numerical and text, even by combining different sources of data with the aim to extract valuable insights on specific topics;
4.  have developed the ability to bring together data exploration, analytical and presentation skills.

The course was taught by both authors in spring 2019, and around 40 students enrolled even though it was optional. The class met twice a week for about two hours each time, and for a total of 20 h. In-class presence was fundamental, though attendance was not mandatory.

Though most students had a degree in political science, the composition of the class varied widely in terms of academic and learning experience, and of cultural and personal background. Around one third of the class was part of Erasmus and Overseas (student exchange programs, respectively within Europe and between Europe and overseas countries). Thereby, students came from many countries including the United States of America (USA), Argentina, Russia, United Kingdom, Germany, Poland, Spain, and Turkey among others. This increased class diversity even further, given the existing differences in the academic education across countries. More importantly, about half of the students were female, a percentage that is in proportion with the ratio of female students in the social sciences.

### 2.1. Technical Tools

As previously pointed out, as instructors, we wanted to lower the entry barriers to social data science, but at the same time provide students with powerful instruments. From this perspective, the choice of the software tool to be adopted for this course becomes of paramount importance as it should reflect the same vision. There are several open source programming languages available, and each one is more appropriate for specific tasks than for others. Of the three languages that were considered preliminary, Python was preferred to R and KNIME for a number of reasons. Indeed, we also considered RapidMiner, based on two reports [14,15] that explained how to integrate RapidMiner and Big Data in a social science class. We then decided to abandon it, as the user-friendliness of RapidMiner was more than offset by the flexibility and power of Python (RapidMiner has nonetheless a module for Python too).

Python is a general-purpose programming language and it has become the most popular language in 2019, as measured by aggregated statistics from several indexes, GitHub repository access and multiple surveys. Albeit powerful, Python is still quite easy to learn. Its simplified syntax with few and simple rules, makes Python one of the most accessible languages. Such simplicity does not affect

other programming qualities, since Python is considered to be reliable, efficient and relatively fast, compared to the other programming languages. Moreover, thanks to its versatility, Python can rely on a broad ecosystem made of a large number of libraries (more than 1400 packages and modules) maintained by a large community of developers.

In addition to developers, Python has also many practitioners. This creates benefits similar to those created by network effects. All levels of Python users, from students to professional programmers, can find support in a massive quantity of resources—from package documentation, to tutorials and professional forums, to GitHub pages—that are all available online. This means that finding prompt and specific solutions to any issue students may encounter while coding is only a Google search away, giving Python an important advantage over less diffused languages. In addition, learning to code in Python, especially when used in the Jupyter Notebook environment, is a valuable skill that can be transferred to many other fields. Thanks to the qualities described above, Python, in fact, has found a broad application in a number of scientific disciplines beyond computer science, such as statistics, physics, biology, and even astronomy.

In the field of computer science, Python is used in a wide range of tasks ranging from building web development frameworks (i.e., Django and Flask), to hardware, to creating video games, and more.

Most importantly for the purpose of this course, Python has become the main programming language used by data scientists overtaking R, which until recently has dominated in this area. It is also widely used for machine learning (ML) and deep learning (DL), which are at the forefront of the latest advancements in the area of AI systems, including natural language processing (NLP) and image recognition, among others. High-tech giants have played a major role in this context by developing the main DL frameworks, namely TensorFlow (Google) and PyTorch (Facebook).

In this course, Python code is executed in Jupyter Notebook, which is a web-based platform that brings together code, data visualization and narrative text. The Jupyter Notebook enables users to perform calculations, write code snippets, show data visualizations, as well as write general text or comments to explain their work. All this can be done interactively in a unique interface, without having to jump from one tool to another. These features make Jupyter Notebook the platform for data science students, and even more so for beginners as it makes it easy to get started. Jupyter Notebook can also benefit teachers in various ways, and for live coding in particular.

Finally, building a collaborative environment by ensuring frequent interaction both between students and professors, and among students, was also considered important. With this in mind a Slack channel was created on the initiative of professors. For those who have never heard of it, Slack is a freemium digital tool that combines the best features of email and online forums. In the context of this class, it was conceived as a virtual space to promote students' interaction by asking questions and providing quick solutions to issues encountered while coding. Thanks to its handy and integrated features, this platform could also serve as a better channel of communication between instructors and students.

### 2.2. Learning Material

The learning material for this class consisted mainly in programming exercises during in-class coding sessions, supported by suggested readings. As additional learning material, students were also strongly encouraged to make use of learning resources available online revolving around the Python ecosystem and data science, such as stack overflow platform, tutorials, technical blog posts, and forums. Making use of the Web for debugging purposes or to find useful scripts/is a common practice for obvious reasons, even among professional coders. Therefore, practicing the ability to find online solutions to code related issues represents another skill to have in the students' skill set. We decided not to release slides and lecture notes until the end of all lessons with the aim to stimulate students to actively engage in live coding in class.

## 2.3. Teaching Structure

Throughout this course, teaching how to code was conceptualized as a top-down approach with most classes focusing on the combination of live demonstration with live coding, where students had to code along with the instructors. Whereas, less importance was dedicated to theoretical lectures.

The first week was devoted to lecture-based classes. Specifically, in the first meeting, instructors provided a detailed presentation of the course content, its structure, learning materials, a first introduction to the software tools, and finally the assessment method. It was followed by a lecture focused on introduction to Big Data and an overview of the most important data sources available online.

The rest of the classes leading up to the last week were dedicated to teaching how to code together with the fundamental data science techniques. During these classes, we performed live demonstrations or live coding, while students were required to code along, rigorously using Jupyter. In particular, as students had no previous programming experience, we started by walking students through the very basics of how to install Python and how to use it in the Jupyter Notebook environment. Python programming syntax was then explained, and more focus was given to coding rules that enable students to work with data. Next, fundamental libraries for data science were introduced and the respective main functions, methods, and attributes were explained using real data sets. While moving from simple to more complex coding exercises, instructors explained data science concepts as they occurred. Throughout the course, we referred to best coding practices and data science techniques, including several algorithms, such as bag-of-words and term frequency-inverse document frequency (TF-IDF) for content text classification, and TextBlob library for sentiment analysis. The part devoted to teaching how to code was quite involved, hence instructors had to be flexible with the schedule to suit students' needs.

During these live coding sessions, various data sets were explored in order to find unexpected patterns and provide insights about real-world issues. Students could also play around with code, trying new functionalities. In this way, they could immediately grasp the power of the software tools they were handling. Just to provide an example, in one particular exercise, instructors went through all the steps and explained the code used to carry out a comparative analysis of the Tweet activity of Donald Trump and Hillay Clinton, main exponents of the Republican Party and Democratic Party respectively, before and after the 2018 midterm elections in the USA with the aim to identify priorities in the political speech and possible differences between them. The accuracy of the results obtained by the text analysis positively surprised students.

## 2.4. Learning Process and Assessment

Students were divided into groups of two, and each group was required to write a short research paper of about eight to ten pages long. The aim of the final paper was to explore and bring new insights into a topic that students care about by leveraging Big Data. To achieve this goal, we asked students to make use of coding skills to handle data as well as apply data science methods to discovering patterns.

Students could choose among any topics of their interest. Leaving complete autonomy on the subject of the paper had a twofold objective: to increase students' motivation on the one hand, and on the other show them that the knowledge and tools offered throughout the course can be applied to a variety of topics in a variety of areas.

Group working in two-person teams was preferred to individual projects so that students working together could help each other by sharing experience and knowledge. Groups also simulate an important aspect of a real professional environment where team building is considered a valuable soft skill to have.

Once students came up with a project-idea, they had to discuss it with the instructors, who assessed whether or not the project would have been feasible. The project idea was basically evaluated in terms of the extent to which data lends itself to cleaning, manipulation, and other methods learned during the course.

By the end of week four, each two-person team was assigned a topic, ensuring that they were not overlapping. We walked the students through the stages of paper preparation and provided suggestions and feedback. Since most groups took advantage of the support, several open meetings were arranged during office hours, but also during class breaks and after class. Help was asked more frequently for issues related to (in order of importance): coding and statistical analysis, and less for the project-idea.

Finally, the last week (two class meetings) was reserved for the presentation of the research projects. Each study group had to give a presentation about the topic of the paper and its structure. This could benefit students in several ways. Students giving the presentation, received additional feedback from instructors about how to further refine and improve the final paper, while improving presentation skills. The other students were invited to ask questions or provide suggestions to their colleagues so that they as well could learn from the discussion of other projects.

Papers were graded based on the combination of the following criteria:

- The originality and creativity of the topic relative to the data set(s) used.
- The extent of data collection, data cleaning, and data preprocessing steps involved in terms of code complexity and appropriateness.
- The breadth and the complexity of the statistical analyses carried out.
- Data visualization techniques where applicable.
- Text analysis where applicable.
- Coding skills and knowledge of the Python ecosystem.
- Writing and presentation skills.
- The ability to extract value from Big Data by providing original insights.

In order to receive additional feedback about the learning experience, two different surveys were distributed to students. The first survey (students' evaluation) is centrally designed and administered by the University of Bologna in a standard format for all degrees. The other one was designed specifically for this class by instructors with the purpose of understanding more about the perceived difficulties encountered by the students throughout the learning process. (The questionnaire is reproduced in Appendix A). Finally, direct communication with students was also used as a source of less formal feedback.

## 3. Results

A total of 12 different research projects were submitted, all of which included the relevant Python code used. Most groups managed to submit their projects by the due date, whereas a few of them concluded it later. Projects covered a broad range of topics, with the most remarkable ones including:

- "Analysis of Amazon reviews of all smartphones with the objective to identify the best smartphone in terms of value for money";
- "Exploring main characteristics of parking fines (specific areas, cars models and timing most likely to be fined) in Los Angeles (LA) urban area, and the possible relationship with the wealth distribution among different areas of LA";
- "Diversity of the political message in the general elections in Italy across the political spectrum, by utilizing social media data";
- "Can human tweets be distinguished from tweets generated by bots?";
- "Can we predict wars?";
- "Degradation vs. requalification: text analysis of how the dailies *Il Resto del Carlino* and *Bologna Today* has been treating this topic with respect to the city center of Bologna".

The other research papers discussed issues related to: Corruption in the Italian public administration; Exploring the relationship between domestic homicides and firearm regulation

among States of the USA; Main features of the funded projects through the crowdfunding platform Kickstarter; Analysis of movie reviews from the IMDB website; City pollution in Barcelona.

### 3.1. Learning

By the end of the course all students were proficient in using Python programming language with the Jupyter Notebook environment to conduct either quantitative data analysis or text analysis, or even both, for the research paper, as demonstrated by the code included in the research projects.

The most encouraging result was the ability of students to apply coding and other skills and data science methods to get insights into a broad range of topics using diverse data sets and data types. More specifically, all students were able to lever the coding skills to handle large amounts of data, both numeric and text, and to manipulate, transform, visualize, and finally, to analyze data by using statistical methods. However, most students showed little knowledge of descriptive statistics, thus, the depth of the statistical analysis was limited.

Nonetheless, the quality of the projects varied widely. Some 5–6 of papers out of 12 were evaluated either with distinction or with the maximum points. Assuming students contributed equally to paper preparation, this means that a total of 10 to 12 students had not only acquired advanced coding skills in Python, but also had gained a deep understanding of the Python ecosystem. To approach their topic, students made use of many different packages available in Python, going beyond those that were covered in class. The scripts used in a couple of research papers were so advanced, in terms of both complexity and combination of library packages, that we expressed some concerns about plagiarism, but such doubts vanished immediately after authors were able to explain the code used. It should be stressed, however, that improving or even maintaining coding skills is a continuous process that requires frequent practicing. This implies that for the student to apply the newly acquired skills in a potential job position in the future, they should keep exercising and further develop their coding ability.

In addition to research projects, the two surveys distributed to students provided some interesting insights into the learning difficulties and other learning aspects as perceived by the students. According to their replies, students seemed aware of the importance of expanding their knowledge by acquiring programming skills. The majority of students appreciated the learning experience overall and found the discussed topics rather interesting, also because no other courses of the Political Science curricula tackled such issues. This was best summarized by one of the students:

> "The course is very interesting, and it allowed me to learn something practical that, otherwise, I would have never studied; the effort to provide students with technical expertise and practical skills is much appreciated."

This perception was further enforced through direct conversations with students. They reported that the decision to attend this course was basically driven by an already existing awareness that coding skills are becoming increasingly useful even in any field of social sciences. Analyzing other survey results, it emerged that, on average, learning programming skills presented a challenge for the students. As expected, the perceived learning difficulties increased as the program moved from basic to more advanced coding topics.

Despite these difficulties, when asked whether or not the students would take the course again should they be able to turn back in time, the majority of the students (78%) claimed they would, while the rest (22%) would not.

Furthermore, when asked what was the most enjoyable part of the entire course, the majority of students (65%) indicated "replicating code by myself", followed by "in-class code examples" for 30% of students, and "project presentations" by only a single student (4% of the total). Interestingly enough, no one selected the "lecture-based classes" option. Clearly, these results should raise serious questions about the traditional way of teaching to code based on lecture-based classes and bottom up approach. Note that the online survey was completed by 23 out of the 40 enrolled students (roughly

60%) as its completion was not compulsory. Therefore, these results may be affected by some degree of self-selection.

*3.2. What Did Not Work*

Despite the positive aspects pointed out in the previous section, there are also other aspects of the program that did not work well. This course expected students to acquire a set of diverse skills. This represented important challenges for many students, even though a large part of the class was able to align with the program.

The results of the official survey indicated that students complained mainly about two aspects, pace of teaching as they found it hard to follow, and the final research project, as some students would have preferred more precise indications on it. Interestingly, students found it hard to learn the exact same topics. This suggests that these do not depend on students' educational background or personal propensity for specific topics rather than others and, thus, are objective difficulties that need to be addressed.

The first critic denotes that students found the number of arguments discussed in the course too profuse. This is the reason why some students were left behind. Reducing class size to a maximum of 10 to 12 students would ensure major attention for each student. Regarding the second critic, explaining the structure of a typical research project in data science by going through each component could help students. The downside of this is that it may limit students' creativity that heavily depends on the amount and type of available data. Students also showed little familiarity with even basic statistical concepts. To make things worse, they were not even aware of such a deficiency.

In terms of the learning process, most students showed low engagement in practicing coding skills at home, especially in the first half of the course. To improve students' motivation, weekly homework and intermediate project assignments should be planned in the program.

Finally, as regards interaction platforms used, students noted that they found it very useful collaborating with each other as they benefited the most from group studies, either in person or online, especially in address coding problems. However, the Slack channel failed to promote interaction among students, though it eased the communication between professors and students. From personal feedback, it emerged that students preferred other traditional digital communication platforms. This might suggest that students felt uncomfortable to share their perceived weaknesses with the entire class.

## 4. Discussion

Over the last decade, the large amount of data that is increasingly being generated, together with the rapid advancements in computer science in general, and NLP techniques, in particular, has opened up new opportunities for the social sciences, including our main concern, i.e., Political Science. Against this context, being equipped with coding skills, which allows students to lever Big Data in the domain of the social sciences has become of crucial importance.

However, the idea of designing a course with the aim of teaching coding skills to students with no previous programming experience to enable them to apply data science methods to the large amount of data in the domain of Political Science, may sound challenging, to say the least. This is because such a learning process involves a steep learning curve, as it requires not only good coding skills, but also deep knowledge of specific domains as well as a touch of creativity. Learning to code starting from zero is hard, and often it can even be frustrating. For beginners, it takes time to reach a level where students can build anything meaningful. Thus, it is easy particularly for students in social sciences with low motivation to quit as they run into apparently insurmountable difficulties.

It is clear, therefore, that finding a teaching format designed in such a way that it stimulates students' interest in learning data science concepts and keeps them engaged throughout the entire course is of paramount importance. This study shows that teaching students how to code by adopting a top-down approach, as opposed to a less effective bottom-up, can help to achieve successful results.

We found that replicating code along with instructors was the most appreciated teaching component by students and theoretical lectures was the least one. In fact, our research shows that students can benefit from live coding sessions and making use of specific tools. The ease of Python programming language and the convenient features of the Jupyter Notebook environment allow students to "get their hands dirty" early in the learning process. This seems to be particularly important because it can help students to not only boost their self-confidence and thus improve their perception of programming abilities, but also keep them engaged even when they cannot make sense of code snippets altogether.

Another result of our research is that going through case studies that apply data science techniques to real-world problems, rather than standard data sets, can increase students' interest and participation during lessons. It appears that solving real world problems can help students shift their focus from the coding process to the general context of Political Science, while at the same time, enable them to see the practical usefulness of these methods.

This is even more true when students are given no restrictions on the data set(s) to work with. The high level of the final research projects demonstrates that the opportunity to work on specific topics students are more interested in can increase their enthusiasm significantly. This is translated into stronger motivation from students to spend more hours practicing coding and learning new things, which, in turn, is the best way to improve their abilities.

A further key component for ensuring successful course outcomes is that instructors should provide students with strong support during the duration of the course (and after), with increasing time-cost. For its own nature, the syntax of any programming language is sensitive to such tiny errors as a missed or added white space or a comma. Especially if students come from a social educational background, it takes months to create these new habits. Hence, having the support of more advanced coders can make the difference. In sum, our research finds that contrary to a somehow widespread belief, students of social sciences are eager to embrace coding skills if they are provided with the right programming tools and to the extent to which these can help them solve problems they care about.

A course based on live coding sessions, however, has also some drawbacks. In-class coding exercises imply that students should be able to follow the arguments explained by the professor, particularly if teaching in classes of relatively reduced size. However, waiting for students to be aligned with instructors can significantly slow down the teaching pace. Therefore, there should be a balance between helping students and completing the course program.

Assuming that graduate students would be comfortable enough with statistics, teaching the fundamentals of applied quantitative statistical methods was beyond the purpose of this course. But as mentioned above, it turned out not to be the case, so we had to spend some time explaining statistics at the expenses of coding.

Clearly, a 10-week course is not enough to teach both statistics and programming, combined with basic data science techniques. To overcome such a problem, it would be necessary for students to be willing to enroll in a quantitative methods course to acquire a more robust knowledge of statistics, as is the case in most graduate programs. Similarly, it would be preferable that students also be familiar at least with the basic syntax of Python. Alternatively, to make sure no one is left behind, laboratory sessions could be introduced, either before or during the course, for those students who find it difficult to keep up with the lessons. This would allow instructors to focus more on the core of programming topics for data science, but it would be more time-consuming and of course require additional resources. In addition, we agree that it would probably be a valuable contribution if at least some time of the class was reserved to consider the epistemological and philosophy of science issues that always arise from any decision to teach one language or method in place of another.

In terms of generalization of results, the diversity of both academic and cultural background of students, together with their total lack of previous programming experience, suggests to us that these results can be achieved by any masters' degree program in Political Sciences. From this perspective, no specific technical tools other than a decent desktop or personal computer and a good internet

connection are required. It would be advisable, however, for instructors to have some experience in working with "real-life" examples, as students show decisively more interest in those.

That said, when it comes to small departments, it may be possible that it is difficult to rely on internal (specialized) staff. There are basically two ways to overcome such an issue. The first is to create synergies between Political Science and other departments of the same university, especially with the computer science departments whenever these are available. In case this is not possible for any reason, the alternative is to rely on external resources. Recruiting an adjunct professor can be a viable solution as they can be hired at no prohibitive cost, thus it is affordable for most departments.

The main limitation of a study such as this one is that it cannot distinguish between the various components of the teaching format such as practical code-based classes, programming tools, communication platforms used, design of the final outcome, project presentations, and the continuous support and feedback provided. This means that we were not able to measure the relative importance of each of these components. Therefore, for a similar teaching format to succeed it would be required that all components were replicated and measured, which we plan to do with more data from future courses.

Finally, it is too early to evaluate the impact on employment for the student cohort on the 2018/2019 academic year and more consistent research should be undertaken in this respect. Albeit definitive evidence is lacking for previous years, as graduates are somehow reluctant to give feedback to the university, even when their job applications are successful, anecdotal evidence at least is quite comforting: between 20% and 30% of graduates who took the "similar" classes in the previous three years confirmed that they were able to secure job interviews in related fields, where the skills learned in class made an impact. Half of them were then offered internships or even full-time positions. Given the usually modest results for Political Science graduates one or two years into the job market, the results have been extremely encouraging for the authors as well as for the department itself.

**Funding:** This research received no external funding.

**Acknowledgments:** The authors would like to thank Emma Michela Giacomello for her help in editing this paper, as well as the three anonymous reviewers for their comments and suggestions.

**Conflicts of Interest:** The authors declare no conflict of interest.

## Appendix A

Survey about the learning experience of the students:

1. In a scale from 0 (less difficult) to 10 (most difficult), please indicate how difficult did you find the main parts covered throughout this class?

- Python Programming
- Pandas DataFrames
- DataVisualization
- Text and Sentiment Analysis

2. Please specify which part of python programming was the hardest to learn?

- Different data types available in python and other object such as lists and dictionaries
- For loops iterations
- Python functions
- List comprehensions
- Other

3.　Please specify which part of data transformation with Pandas was the most difficult for you?

- Reading files from .csv, .json formats and eventually save them
- Processing data
- Indexing by row(s) and/or column(s) with .loc, .iloc
- Selecting part of DataFrames based on logical conditions

4.　Please specify which part of text analysis was the most difficult for you?

- Pre-processing steps including tokenization removing stopwords and similar
- Getting the most common words (bag of words)
- TF-IDF algorithm

5.　What did you enjoy the most during this course?

- Lecture-based classes
- Code examples
- Replicating code by yourself
- Project presentations
- Nothing at all

6.　If you could turn back in time, would you take this course again?

- Yes
- No

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
