# Peer review of "The “Social” Side of Big Data: Teaching BD Analytics to Political Science Students"

_2504-2289, doi:10.3390/bdcc4020013_

Round 1
Reviewer 1 Report
This version of the manuscript is substantially improved and more interesting than the first version. The contribution is no longer overstated, and the focus is on what the course actually does as well as thoughtful discussion of the curriculum's strengths and weaknesses.
I think that it would benefit from another editing round by the authors (for example, a 'not' is missing in the abstract that seems very important!), and it definitely needs copy editing by a native english speaker.
Author Response
We did an overall checking and adjust of the text (which did need improvement) and had the paper proof-read by a native English speaker. We think that the overall result is a greater improvement of the whole paper.
Reviewer 2 Report
I have no further comments to add as the authors addressed most of the original suggestions for revision.
Author Response
We would like to thank the reviewer for his previous comments and suggestions
Reviewer 3 Report
An overall impression: The interesting paper discussing the interesting teaching experience. Thus, all in all it is worth publishing and sharing with others. The most valuable part seems to be the selection of the teaching tools, data sets and the overall course strategy. The encountered problems and difficulties pointed out by the authors are also important as they may suggest the proper design of the similar curriculum.
To be improved: The paper needs significant improvements in English (sentence structure, grammar, use of words, spelling) and the flow of narrative. In Introduction the authors seem to get off the main topic in few places making the narrative difficult to follow. It seems that the clear statement how critical it is to understand the impact of Big Data and analytics on democracy is missing. Otherwise, why we would teach students these skills?
More detailed comments attached.

Author Response
We substantially revised and adjusted the whole text according to what the first reviewers indicated, and had it proof-read by a native English speaker. More specifically, we include all the changes that were highlighted by the third reviewer in the enclosed Pdf file.
Albeit extremely important, we decided, however, not to discuss in details the issue of teaching Big Data to democracy. Because it is a very important topic it would have required a much more extensive discussion, which would have increase the size of the paper. But we indicated in the paper that some attention should be dedicated to the philosophy of science issue that may arise from deciding to teach something instead of something else.
As for the point that Python has many self evident advantages, while we fully agree, this point is not so easy to grasp for other colleagues. For example in our department there's an ongoing discussion of the advantages/disadvantages of Python and R. R is not a programming language, it cannot do as many tasks as Python, but we are still engaged with many colleagues on this point. Thus we felt that it was important to use the paper to illustrate once again the many overall advantages of learning a 'true' programming language instead of, say, a statistics package as R.
Round 2
Reviewer 3 Report
Thank you for revising the paper. It can be accepted as is.
This manuscript is a resubmission of an earlier submission. The following is a list of the peer review reports and author responses from that submission.
Round 1
Reviewer 1 Report
This manuscript describes a course where students learn the python programming language and complete an 8-10 page project. The premise is that social science students can benefit from these skills but they are not being taught.
The premise is not correct. There are many undergraduate data science programs geared to social science students, and the number grows every year.
Just two of many examples:
https://statistics.stanford.edu/data-science-minor
https://lpsonline.sas.upenn.edu/academics/bachelors-degree/baas-concentrations/data-analytics-and-social-sciences
Moreover, the syllabi for the classes in those programs can often be viewed, providing considerably more detail than what is provided in this article.
I commend the authors for providing this valuable service to their students, but the article is not ahead of the curve as the authors suggest.
Reviewer 2 Report
I think this is an interesting article that explores an important pedagogical topic in political science. I mostly found it interesting, but there are several substantial revisions I would recommend to the authors.
I believe the authors oversell the novelty of this study a bit by only referencing one pedagogy article from political science that explores data analysis and visualization. Since that article both references and is referenced by other authors within the same journal exploring these topics, it would be worth expanding the reference list. Subdiciplinary journals like International Studies Perspectives also include pedagogy articles and may be worth looking at. Depending on the journal's limitations for space and reproduction policies on visual materials, it would be interesting to see more feedback from the student evaluations and perhaps some presented visually. If the evaluations provided some student comments, it may be useful to include these. It seems we were only told the results of 1-2 questions and this data was presented without a lot of nuance, so I was interested in knowing more. Related to this, I would like to see the authors reflect a bit more on what didn't work. Since only 1/3 of the students seemed to have passed with distinction, this is far from an unqualified success. The author's reflection on the struggles of 2/3 of the course mostly center on ideas that the students were inadequately prepared and didn't like using Slack. It seemed unsatisfying that the instructors didn't seem at all reflective about how they themselves might have done things differently that led to student success. Some comment on the replicability of a course like this might be useful. One thing that came to my mind, which was not addressed in this article, is that many departments/instructors likely don't cover big data analytics because so few political science faculty are themselves proficient in this area. My suspicions are that a course such as this would be difficult to implement in departments that are small and/or can't offer this as part of a methods sequence (with students taking a basic research methods course first). It would be interesting to see the authors comment more on how widely an approach like this could be implemented.
As a couple of final notes, the article requires editing for English language. Grammatical and spelling errors occur throughout the text. Should this be published (and I think it is certainly a subject worth publishing on, if shortcomings are addressed), the authors may also wish to consider hosting sample materials and a syllabus as an online appendix or on their personal websites with a link in the text, for those who wish to explore their approach further.